# Lifestyle Intervention on Body Weight and Physical Activity in Patients with Breast Cancer Can Reduce the Risk of Death in Obese Women: The EMILI Study

**DOI:** 10.3390/cancers12071709

**Published:** 2020-06-27

**Authors:** Laura Cortesi, Federica Sebastiani, Anna Iannone, Luigi Marcheselli, Marta Venturelli, Claudia Piombino, Angela Toss, Massimo Federico

**Affiliations:** 1Department of Oncology and Hematology, Azienda Ospedaliero-Universitaria di Modena, 41124 Modena, Italy; federica.sebastiani@gmail.com (F.S.); lmarcheselli@unimore.it (L.M.); martaventurelli@msn.com (M.V.); Claudia.piombino@outlook.com (C.P.); angela.toss@unimore.it (A.T.); mfederico@unimore.it (M.F.); 2Department of Surgery, Medicine, Dentistry and Morphological Sciences with Transplant Surgery, Oncology and Regenerative Medicine Relevance, University of Modena and Reggio Emilia, 41124 Modena, Italy; anna.iannone@unimore.it

**Keywords:** obesity, BMI loss, breast cancer, diet, physical activity, overall survival

## Abstract

Background obesity and sedentary lifestyle have been shown to negatively affect survival in breast cancer (BC). The purpose of this study was to test the efficacy of a lifestyle intervention on body mass index (BMI) and physical activity (PA) levels among BC survivors in Modena, Italy, in order to show an outcome improvement in obese and overweight patients. Methods: This study is a single-arm experimental design, conducted between November 2009 and May 2016 on 430 women affected by BC. Weight, BMI, and PA were assessed at baseline, at 12 months, and at the end of the study. Survival curves were estimated among normal, overweight, and obese patients. Results: Mean BMI decreased from baseline to the end of the study was equal to 2.9% (*p* = 0.065) in overweight patients and 3.3% in obese patients (*p* = 0.048). Mean PA increase from baseline to the end of the study was equal to 125% (*p* < 0.001) in normal patients, 200% (*p* < 0.001) in overweight patients and 100% (*p* < 0.001) in obese patients. After 70 months of follow-up, the 5-year overall survival (OS) rate was 96%, 96%, and 93%, respectively in normal, obese, and overweight patients. Overweight patients had significantly worse OS than normal ones (HR = 3.69, 95%CI = 1.82–4.53 *p* = 0.027) whereas no statistically significant differences were seen between obese and normal patients (HR 2.45, 95%CI = 0.68–8.78, *p* = 0.169). Conclusions: A lifestyle intervention can lead to clinically meaningful weight loss and increase PA in patients with BC. These results could contribute to improving the OS in obese patients compared to overweight ones.

## 1. Introduction

Breast cancer (BC) is the most common cancer in women worldwide. In Italy, between 2008 and 2012, breast cancer reached an age-standardized incidence rate of 126.9 per 100,000 women in northern Italy, 111.2 per 100,000 women in central Italy, and 98.9 per 100,000 women in southern Italy, Sicily, and Sardinia [1]. Since BC prognosis has significantly improved in developed countries due to earlier diagnosis and treatment improvements, BC survivors comprise the largest group of cancer survivors, with a prevalent number of cases equal to 799,196 in Italy. The five-year survival rate is equal to 87%; this rate is higher than both the European average (81.8%) and the North European average (84.7%) [2]. However, BC outcome is different according to histological subtypes, defined by three broad biological subtypes, based on the expression of diagnostic biomarkers estrogen (ER) and progesterone (PR) receptor-positive; HER2 positive; and triple-negative (lacking hormone receptors and HER2). A further stratification of the disease includes six intrinsic subtypes (normal, claudin-low, luminals A and B, HER2 enriched and basal) [3] and four triple-negative molecular subtypes (basal-like 1, basal-like 2, mesenchymal and luminal androgen receptor) [4].

It is well known that obesity influences BC risk and affects survival in patients diagnosed with BC. For this reason, in a recent study, we evaluated the relationship between body mass index (BMI) and the risk of breast cancer (BC) as well as the outcome in 14,684 women recruited between ages 55 and 69 who resulted eligible to participate in the mammography screening program (MSP) in the province of Modena, Italy. We reported a significantly higher risk of BC in obese women and we also demonstrated that obese women had more second events and poorer event-free survival compared to non-obese women [5].

Already in 2010, a review of 43 prospective studies evaluating the effect of obesity on BC prognosis had shown a poorer overall and BC-specific survival in obese patients compared to non-obese women [6]. Two further meta-analyses have linked obesity and being overweight to a higher risk of recurrence and mortality [7,8]. A recent paper summarizes all studies on obesity and breast cancers showing an association with a 35% to 40% increased risk of breast cancer recurrence and death, thus resulting in poorer survival outcomes. This is most clearly established for estrogen receptor-positive breast cancer, with the relationship in triple-negative and human epidermal growth factor receptor 2–positive subtypes less well established. [9]. Also, weight gain after BC diagnosis has been associated with poorer prognosis [10].

On the other hand, physical activity (PA) has been associated with improved outcomes in BC survivors in many epidemiological studies [11,12,13,14,15,16,17,18], and meta-analysis confirmed that not only can PA reduce BC mortality, but it can also reduce several causes of mortality [19,20,21,22].

International recommendations for cancer prevention published in 2007 by the World Cancer Research Fund, include eating a healthy diet, being physically active and maintaining a healthy weight [23]. In the 2014 update report on BC survivors, the panel of experts concluded that evidence is still not strong enough to make specific recommendations for women with BC [24]. Nevertheless, there are indications of connections between better survival after BC and a healthy body weight, being physically active, eating foods containing high levels of fiber, eating foods containing soy, and including a lower intake of total fat in particular saturated fat in one’s diet [25].

Findings from experimental studies involving BC survivors suggest that a behavioral intervention may support clinically meaningful weight loss [26,27,28,29,30].

However, in our knowledge, no studies have reported the effects of lifestyle interventions on body weight in breast cancer survivors in the Italian population so far.

The purpose of this single-arm pilot study was to test the feasibility and efficacy of a lifestyle intervention through diet and PA on body weight in a population of more than 400 patients with BC followed at Modena Cancer Center, Modena, Italy.

## 2. Results

A total of 430 patients were included in the study. Fifty-seven patients developed a second tumor (34 BCs and 23 other cancers). The time of study enrollment was equal to 78 months. Ninety-two patients left the study before the end (21.2%). On 31st March 2020, excluding seven patients who left the study during the first year, the median follow-up was equal to 70 months (range 12–124). The median follow-up was equal to 64 months in the underweight patients, 71 months in the normal weight group, 71 months in the overweight patients, and 70 months in the obese patients. The median time on treatment in underweight patients was 7 months, in normal weight patients was 84 months, in overweight patients was 64 months and in obese patients was 96 months. The median age at the diagnosis was 53 years (range 30–76 years). We observed that, considering BMI values, 3 (0.7%) patients were underweight at the baseline, 100 (23.3%) had normal weight, 167 (38.8%) were overweight, and 160 (37.2%) were obese. In total, 267 patients (76%) were overweight and obese. At the last follow-up, 127 patients were undergoing hormonal treatment. Baseline characteristics of the study population are reported in Table 1. No differences were found in disease stage and type of therapies (chemotherapy alone, hormonal therapy alone, or combined treatments) among patient groups. When the only hormonal treatment was evaluated, aromatase inhibitors (AI) were the most frequently administered therapy, followed by tamoxifene, and lastly by a combination treatment. Normal weight patients received tamoxifene more frequently, whereas obese patients were mostly treated with AI (*p* = 0.041). On the whole, with regards to the menopausal status, the vast majority of patients were in postmenopausal condition (49.8%); this hormonal status was particularly predominant in obese women (56.9%) and overweight patients (52.1%), whilst 58% of normal weight patients were in premenopausal status (*p* < 0.001). Luminal cancers were the most frequent tumor subtype, being luminal A equal to 48.7% in the obese group and luminal B equal to 36.4% in the overweight group. Triple negative tumors arose more frequently in the normal weight group of patients (9%) (*p* = 0.001). A previous BC was particularly evidenced in overweight patients compared to other groups (*p* < 0.001). Finally, looking at tumor size and grade of invasive BC, obese patients showed the highest percentage of tumor larger than 5 cm (30%) (*p* = 0.001) and the highest percentage of grade III (60%) (*p <* 0.001).

### 2.1. Weight, BMI, and Physical Activity

In regard to the definitive results of the lifestyle intervention, only patients belonging to the overweight and obese groups derived a benefit. Any change in BMI was registered for underweight or normal-weight patients. As shown in Figure 1A, we reported a statistically significant BMI loss in the overweight patients from the beginning to the first year and in the obese group, from the beginning to the end of the study, whereas no statistically significant differences were seen in the underweight and normal-weight patients. Globally the median weight at the baseline was equal to 74.1 (212 ± 20,6) kg and 70.4 (168 ± 18.3) kg at the last follow-up, with a statistically significant decrease of 5.5% (*p* < 0.001, mean difference −3.39, 95%CI = −4.41−2.36).). The differences in weight throughout the study period reflected the BMI changes amongst the four patients’ categories (Figure 1B). Particularly obese women had a weight reduction of 2.8 kg, moving from 86.8 to 84 kg (3.2%, *p* = 0.048, mean difference −1.51, 95%CI = −1.80−0.01). Overweight women moved from 72.2 kg, at the entry to 67.6 after one year (−6.3% *p* < 0.001, mean difference −1.90, 95%CI = −3.10−0.70). In parallel, there were also highly significant (*p <* 0.001, mean difference 6.7, 95%CI =+ 5.6 + 7.8) changes in total physical activity levels. Normal patients increased their weekly activity by 1 h in the first year, from 1.0 to 2.25 at the end of the study with an increment of 125% (*p* < 0.001, mean difference 1.83, 95%CI= + 1.6 + 2.1), overweight patients increased physical activity moving from 0. to 2.0 h throughout the entire study period (increment of 200%) (*p* < 0.001, mean difference 95%CI= + 1.7+ 2.3) finally obese patients moved from 0 h per week to 1.0 throughout the study period (increment of 100%) (*p* < 0.001,mean difference 1.47 95%CI = +1.1 + 1.8).

In Figure 2, differences in physical activity modification are shown for each group of patients. Either overweight or obese women show the most important changes in physical activity during the first year, with a slight decrease at the end of the study.

In the overweight group the benefit of the intervention was more effective at 12 months, and less at the end of follow-up. In the obese group of patients there was a continuous benefit along the follow-up period time.

### 2.2. Survival Curves

Breast cancer recurrences including local and distant metastases occurred in 55 patients (12.8%). There were 38 (9.0%) patients who died from BC recurrences (24 patients) or other causes (14 patients). Excluding patients with a follow-up less than 12 months and patients with a second tumor, the 5-year OS was equal to 95%, with 35 deaths, and the 5-year PFS corresponded to 90%, with 55 relapses (Figure 3A,B). In regard to the PFS, no statistically significant differences were seen among normal plus underweight patients versus overweight and obese patients (94 vs. 93 vs. 93 months, respectively) (Figure 4B). The association between overweight and obese women compared to normal and underweight women was not statistically significant (*p* = 0.097).

Other causes of death were represented, in three cases by other cancers, in four cases by pneumonia disease whereas in the remaining seven cases the cause of death was unknown. No differences in BC specific survival were seen between overweight and normal patients (data not shown). Since only 3 patients were underweight, they were combined with the normal weight patients; all together were compared to overweight and obese patients, having a statistically significant better OS (96 vs. 93 months, *p* = 0.027, HR 3.69, 95%CI = 1.17−13.4)) than overweight patients, but not compared with obese patients (96 vs. 96 months, *p* = 0.169, HR 2.45, 95%CI = 0.68–8.78)) (Figure 4A). By associating overweight and obese patients, the OS comparison was equal to 96 months for underweight plus normal-weight patients and 94 months for overweight plus obese patients (*p* = 0.051, CI = 1.00–6.40). In regard to the PFS, no statistically significant differences were seen among normal plus underweight patients versus overweight and obese patients (94 vs. 93 vs. 93 months, respectively). (Figure 4B). The association between overweight and obese women compared to normal and underweight women was not statistically significant (*p* = 0.097).

Table 2 describes survival analysis of weight and other characteristics by Cox’s proportional hazards regression for univariate analysis. No statistically significant differences were seen in OS and PFS for BMI, menopausal status, hormonal treatment, and tumor phenotype, although a worse prognosis was shown for overweight BMI, postmenopausal status, aromatase inhibitors treatment, and luminalB/luminal HER2 subtype. The only factor considered as statistically significant in univariate and multivariate analyses was a previous BC diagnosis (OS:HR = 2.78, *p* = 0.007; PFS:HR = 3.10, *p* = 0.002).

A multivariate analysis among BMI, stage, and second tumors were performed in order to evidence if some factors were independently related to the overall survival in specific subgroups of patients (Table 3). Overweight patients, but not obese ones, BC diagnosed at stages III and IV, the presence of second tumors, and tumor size > 2 cm were independent risk factors of death

The percentage of second tumor in normal plus underweight, overweight and obese patients was 2%, 5%, and 6% (*p* = 0.431).

## 3. Discussion

Weight is becoming a major health issue in the USA with >60% of American adults being obese and overweight [31]. An association between risk of occurrence and obesity has previously been reported for various cancers. Recently, numerous studies including multivariate analyses demonstrated an independent prognosis effect of obesity on the risk of BC recurrence, PFS, and OS [5,6,32]. For example, a retrospective analysis of the Danish Breast Cancer Cooperative Group (DBCG) database including 18,967 BC patients, treated between 1977 and 2006, revealed that overweight or obese patients more often presented at diagnosis with features associated to poor prognosis. Even after adjusting for classical prognostic factors, obesity remained an independent risk factor for the development of disease metastases and for BC related death [33]. Similarly, a retrospective multivariate analysis of 2887 node-positive BC patients enrolled in the BIG 02-98 trials showed that obesity remained an independent prognostic factor for both PFS and OS despite more aggressive tumor characteristics in obese patients compared to non-obese counterparts at diagnosis [34].

The results of our study show that the rate of obese and overweight BC women was very high in Italian population (76%), despite following the Mediterranean diet [35]. Therefore, we investigated the feasibility and efficacy of a lifestyle intervention through diet and PA on body weight and therefore prognosis of 430 patients with BC followed at Modena Cancer Center, Modena, Italy. No differences in disease stage were found according to BMI, although obese patients showed the highest percentage of tumors larger than 5 cm and grade III, as well as demonstrated by several studies in which obese women develop significantly larger sizes or advanced BC compared to normal weight women [36], Our multivariable analysis confirms that being overweight patients is an independent risk factor of death, such as having an advanced stage at diagnosis (stage III and IV),and a tumor size larger than 2 cm, regardless of the weight. However, the increased number of second tumors in different BMI categories do not seem to have an impact on BC survival.

The relationship between obese patients and AI treatment is well known, but we have clearly shown that obesity is a metabolic condition regardless the hormonal treatment and it is not the consequence of the therapy. In fact, obesity represents the most frequent condition in postmenopausal status, where AI are mostly used. However, as reported by a recent study that investigated the BC prognosis in obese women treated with AI [37], no worse outcome was seen compared to normal patients, not supporting the speculation that an elevated aromatase activity decreases the clinical efficacy of AIs in these women [38,39,40,41,42]. Our BC specific survival curves did not show a difference between overweight and normal patients, suggesting that other causes of death could impact on the worst prognosis in this group of women. In fact, the other causes of death were affecting the pulmonary district; it was particularly evident both in the obese group of patients (one died of acute respiratory distress syndrome, one of chronic obstructive pulmonary disease) and in the overweight group (the death was due to pneumonia disease in two cases). These disease conditions have been related to in increased body weight and a recent study has shown that a regular physical activity in overweight and obese patients is associated to a lower risk of developing those morbidities [43]

We have shown a relationship between obese patients and luminal A subtype tumor, as already seen in another study [44]. The first objective of our research was reached and we demonstrated that, after ten years from the beginning of the study, healthy lifestyle guidelines provided by a dietician who lays out a personal program of nutrition education and PA is able to significantly reduce BMI and increase weekly PA in overweight and obese women. Moreover, obese women continue to decrease their weight and BMI after medical monitoring.

Furthermore, we documented a statistically significant worse prognosis of overweight patients compared to obese and normal-weight patients. By the univariate analysis, no statistically significant differences were seen among BMI, menopausal status, or tumor phenotypes and survival curves, although a trend toward worse prognosis was revealed in overweight patients. Since the study was concluded after seven years, we can’t derive any conclusion on the better prognosis of obese patients than overweight patients at the end of data collection. However, a longer intervention control and data registration could provide new insight on efficacy of this program. We are conscious that a benefit on other diseases, such as cardiovascular risk factors, or diabetes is probably seen with a BMI decrease in the range of 5–10%, being considered as secondary objective in a large randomized trial for adults with a recent diagnosis of type 2 diabetes [45], but it remains unclear whether a BMI decrease less than 5% can impact on BC prognosis

The strength of this research derives from the methodology used to provide a real and effective lifestyle intervention in BC patients, able to statistically reduce body weight, particularly in overweight and obese women throughout the study period. A better prognosis in obese than in overweight women was seen, in contrast to the vast majority of studies where obesity represented the worst category of patients in terms of death and recurrence [46].

The limit is represented by the short period of follow-up, knowing that recurrences and death in patients with hormonal positivity tumors arise after the first five years from the diagnosis. This could have reduced the real impact of the lifestyle intervention on obese women, who show a significant yet slight decrease in weight and BMI. Probably an extension of the study would continue to improve health conditions of our patients.

Another important limitation is the high drop-out rate from the study, particularly in the first year of intervention; a high rate in drop-out could probably be reduced by more frequent reminders to patients who do not obtain a BMI decrease at the first six month visit

## 4. Materials and Methods

### 4.1. Methods

The study was approved by the Ethic Committee of Modena, Italy. It is a single-arm experimental design conducted to test the efficacy of a lifestyle intervention in patients with BC treated at Modena Cancer Center from November 2009 to May 2016. All participants provided written informed consent in accordance with the Declaration of Helsinki.

### 4.2. Eligibility Criteria

The study population included 430 women. We included patients with a new diagnosis of hormone positive breast cancer, treated by surgery, with or without radiotherapy, hormonal therapy, and/or chemotherapy in adjuvant setting.

### 4.3. Study Procedures

At the first visit, women were weighed and height was measured; arm, waist, and hip circumferences, along with the tricipital, bicipital, subscapular, and suprailiac folds were also measured; BMI, defined as weight in kilograms divided by height in meters squared (kg/m^2^). Based on the BMI, women were grouped into the weight categories recommended by the World Health Organization (WHO) [46]: underweight (BMI < 18.5 kg/m^2^), normal-weight (BMI = 18.5–24.9 kg/m^2^), overweight (BMI = 25–29.9 kg/m^2^) and obese (BMI ≥ 30 kg/m^2^). Nutritional history and physical activity (hours/week) were inquired. Finally, the dietician provided patients with recommendations for a healthy lifestyle and a survey on their health status.

One month later, following the evaluation of the survey on health status, the dietician provided patients with a personalized program of nutrition education and physical activity. Written resources containing the food pyramid, healthy eating guidelines and tips, basic information on portion size, examples to reduce food intake (maximum calorie reduction, 500 kcal/d), and dietary recommendations were delivered to patients. These latter were specifically designed and tailored taking into account nutritional needs, comorbidities, and other information and/or needs communicated during the first meeting and consisted of daily and weekly meal plans and suggestions for meal composition stimulating patients to self-arrange their diet according to the MedDiet. Patients were re-evaluated at 6, 12 months, and at the end of follow-up. During the subsequent meetings, the dietician also evaluated the adherence to the dietary recommendations, made eventual adjustments, assessed anthropometry, and assisted participants in maintaining motivations.

### 4.4. Definition of BC Subtypes

Her2/neu testing was carried out at a single pathology laboratory in Modena by immunohistochemistry, and the results were scored as follows: 0, 1 = negative, 2 = indeterminate, and 3 = positive. Patients with HER2 test results reported as “indeterminate” were evaluated by fluorescence in situ hybridization (FISH). ER and PgR testing were conducted with a single report format of “positive” or “negative” test results, as measured by immunohistochemical analysis (clone 6F11, Ventana, for ER; and clone 1E2, Ventana, for PgR) and staining by Ventana Benchmark autostainer. ER and PgR receptor status were tested by evaluating the percentage of nuclear immunoreactivity with respect to all the nuclei in the neoplastic cells, independently of the staining intensity. Nuclear staining 10% of either ER or PgR was considered a positive result. Ki-67 labeling index was determined with the MIB1 monoclonal antibody as nuclear immunoreactivity. The cut-off was equal to 14% to subdivide luminal A and luminal B tumor. Luminal A were tumors with ER/PgR/HER2-/Ki67 < 14%, luminal B were tumors with ER/PgR/HER2-/Ki67 ≥ 14%, luminal/HER2 enriched were tumors with ER/PgR/HER2, triple-negative tumors were ER-/PgR-/HER2-, and HER2 were tumors with ER-/PgR-/HER2.

### 4.5. Dietary Intervention

The dietary intervention was based on the MedDiet which was first introduced and described by Keys et al. in “The Seven Countries Study”. Consequently, several other studies confirmed the beneficial effects and outcomes of the MedDiet on health mainly reducing mortality, cardiovascular disease risk factors, and cancer [47].

The MedDiet is based on a balanced use of foods rich in fiber, antioxidants, and unsaturated fats, a healthy approach designed to reduce the consumption of animal fats and cholesterol in a diet with an appropriate balance between energy intake and expenditure [35]. Therefore, the MedDiet is characterized by high consumption of vegetables, fruits, non-refined cereals, legumes and potatoes, moderate consumption of fish and poultry and low consumption of full-fat dairies, red meat and derived products, as well as homemade baked goods. Olive oil is the basic source of fat used for food preparation and condiment. Meals are often accompanied by low-to-moderate amounts of wine. The relationships between the macronutrient in the MedDiet is 55–60% of carbohydrates of which 80% complex carbohydrates (bread, pasta, rice), 10–15% of proteins about 60% of animal origin (especially white meat, fish), 25–30% fat (mostly olive oil) [35].

The MedDiet is typically represented graphically in the shape of a pyramid whose most updated version is described by Bach-Faig et al. [48]. The graphic representation follows the previous pattern: at the base, food items that should sustain the diet and provide the highest energy intake, and at the upper levels, foods to be eaten in moderate amounts such as animal source foods and foods which are high in sugars and fats that should be eaten in moderation or occasionally [48].

Along with recommendations regarding the proportion and frequency of food consumption, the incorporation of cultural and lifestyle elements is one of the innovations of the latest version of the pyramid. These concepts are represented outside of the pyramid, and at its base the following concepts are represented: moderation, socialization, culinary activities, physical activity, adequate rest, seasonality and tradition, local/homegrown products, eco-friendly and biodiverse products.

Eating habits were recorded using a food diary. A food diary dietary assessment approach records consumption of food and beverages as they are consumed throughout the reporting day thus making it more accurate than food frequency questionnaires which depend on retrospective recall. We recommended a period of 7 days for record-keeping in the week leading up to follow up appointment. For checks after 3, 6, and 12 months, food diaries were recorded over a period of 7 days every month. Food diaries allow the patient to record all food intake in the established period, after which the nutrient intake may be calculated and averaged. At the follow-up visits, we were able to address the compliance to the diet by food diary registration

### 4.6. Physical Activity Program

As far as physical activity is concerned, the counseling approach was mainly used to stimulate patients to increase their physical activity. A motivational method instead of a more common prescriptive method was used to favor a more active lifestyle taking an incremental approach i.e., trying to increase the general level of activity throughout the day and every day up to a sustainable and compatible level with physical conditions and available time. A physical activity diary, containing the type and duration of exercises, was offered in order to register the patient’s perception and comments every time. The aim was to maintain a regular activity over time or to provide an instrument for modifying some exercises in order to become more pleasant. A self-esteem evaluation was also registered and measured to reach a high level.

During the meetings, patients and a dietician tried to set achievable goals tailored to the physical conditions of every single patient. People with (very) limited physical activity levels were stimulated to start identifying and ensuring ways to increase their physical activity. For example, obese patients were stimulated to identify daily activities that could increase their physical activity (e.g., walking, riding a bike, getting off the bus one stop before, walking up the stairs instead of taking the elevator, walking the dog, walking with their children, etc.). All patients without any physical constrain were stimulated to identify and take up/maintain regular physical activity. For these latter patients, the final goal was a habitual physical activity workload > 20 METs/h (where 1 MET is equal to 3.5 mL of oxygen spent for kg of weight per hour) in a week (corresponding to 3-h/wk moderate-intense physical activity) which was used in the study of Montesi et al. [49].

At the follow-up visits, patients exhibited the physical activity diary; the daily type and duration of exercises were registered in order to verify the compliance to the personal program.

Patients who continued in the established program were satisfied with the results, in terms of behavioral changes. Many of them tried to convince all family members to change their lifestyle, according to the dietary intervention and physical activity program. On the contrary, patients who did not experience a change in weight and their BMI felt inadequate.

### 4.7. Statistical Analysis

A database was set up at our institution, which consisted of collecting individual details, weight, height, BMI, physical activity, and follow-up data, using the Excel program by Window version 10. The follow-up was calculated from the date of the BC onset to the endpoint of interest: the date of death or the end of the study period. The ANOVA test was used to determine differences in clinicopathological features between groups. Since the BMI, Kg, and PA were measured at different time points over the same patient, we checked the association of those variables as function of time points by means of ANOVA for repeated measure. We reported a raw *p*-value, not adjusted by multiple comparison. All statistical tests were two-sided. *p* < 0.05 was considered statistically significant, and the *p*-value was two-sided for all analyses. Survival curves were estimated using the Kaplan-Meier method including the log-rank test group comparison. Univariate and multivariate analyses of PFS and OS were conducted using a proportional hazards Cox regression model. All analyses were performed using the Stata statistical package (version 10 SE).

## 5. Conclusions

A lifestyle intervention can lead to clinically meaningful weight loss and increase PA in patients with BC. Our obese patients showed a better OS than overweight patients. The worst prognosis of overweight/obese patients aged over 55 years is likely related to death from other causes.

## Figures and Tables

**Figure 1 cancers-12-01709-f001:**
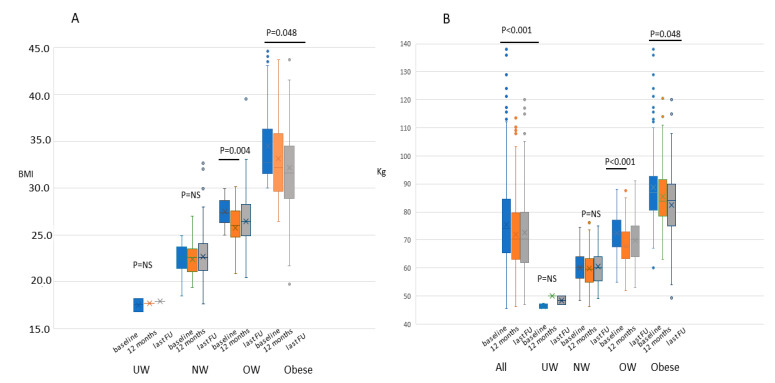
Figure 1 represents the BMI (**A**) and Kg (**B**) changes for patients categories’ throughout the study. (**A**) represents the body mass index (BMI) evaluation at the baseline (blue), at 12 months (orange) and at the end of the study (grey) in the underweight (UW), normal-weight (NW), overweight (OW) and obese patients by the whisker plot. The dots represent the outlier values. The estimated difference between BMI, weight, and physical activity along time-points was performed by means of ANOVA with repeated measures. No statistically significant differences (*p* = NS) were seen in the UW and NW groups along the time, whereas a statistically significant difference was shown in OW patients after one year from the baseline (*p* = 0.004, mean difference −0.72, 95%CI = −1.20−0.23), but not at the end of the study (*p* = 0.065, mean difference −0.45,95%CI = −1.1−0.54), and in obese patients (*p* = 0.048, mean difference −0.92, 95%CI = −1.82−0.01) at the end of the study, with a slow but progressive decrease of BMI. (**B**) represents weight evaluation at the baseline (blue), at 12 months (orange) and at the end of the study (gray) in the total population (all), underweight (UW), normal-weight (NW), overweight (OW) and obese patients by the whisker plot. The dots represent the outlier values. Globally the median weight decreased from 74.1 to 70.4 kg at the end of the study (*p* < 0.001 mean difference −3.39, 95%CI = −4.41−2.36), obese women moved from 86.8 to 84 kg (−3.2%, *p* = 0.04 8, mean difference −1.51, 95%CI =−1.80 −0.01), overweight women moved from 72.2 kg, at the entry to 67.6 after one year (−6.3% *p* < 0.001 mean difference −1.90, 95%CI = −3.10−0.70). No differences throughout the study period were seen for normal and underweight patients (*p* = NS).

**Figure 2 cancers-12-01709-f002:**
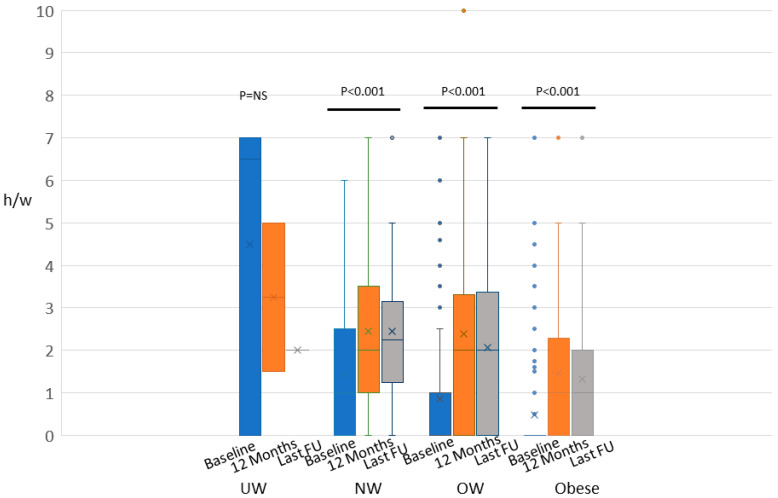
The figure represents the physical activity as hours per week (h/w) evaluation at the baseline (blue), at 12 months (orange) and at the end of the study (gray) in underweight (UW), normal weight (NW), overweight (OW) and obese patients by the whisker plot. The dots represent the outlier values. The estimated difference in physical activity along time-points was performed by means of ANOVA with repeated measures. No statistically significant differences (*p* = NS) were seen in the UW group, whereas in the NW, OW, and obese groups, a statistically significant difference was shown (*p* < 0.001).

**Figure 3 cancers-12-01709-f003:**
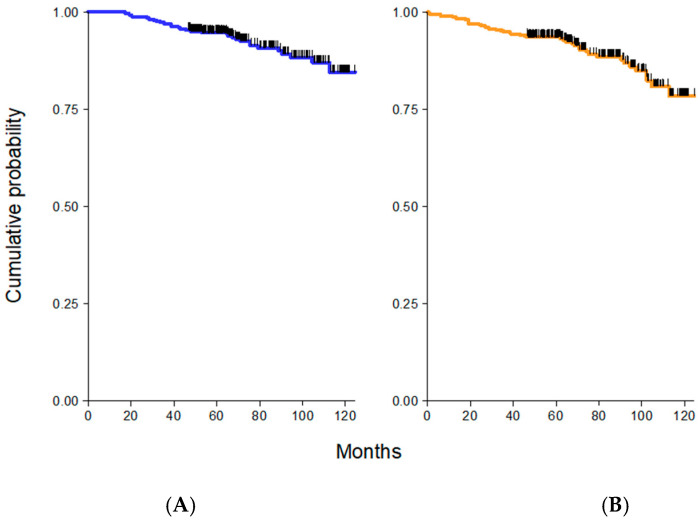
Five-year overall survival (OS), excluding patients with follow-up less than 12 months and with second tumors, was equal to 95% (**A**) and five-years progression free survival (PFS) was equal to 90% (**B**).

**Figure 4 cancers-12-01709-f004:**
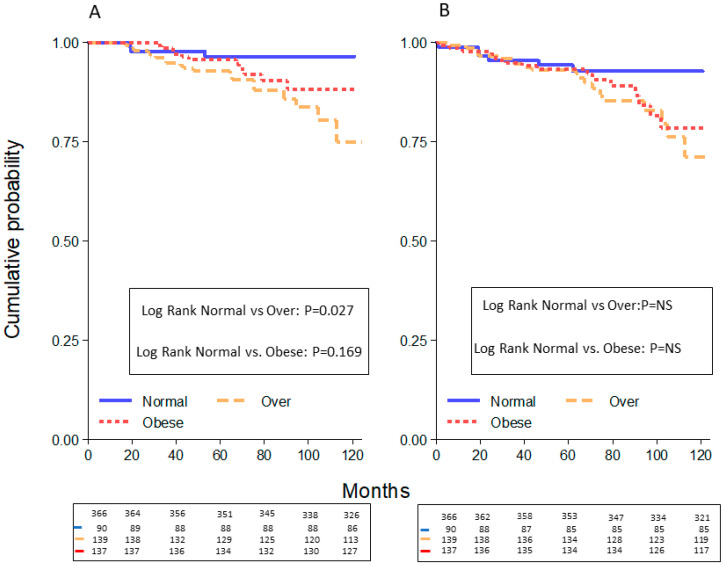
The five-year overall survival (OS) and progression free survival (PFS), excluding patients with follow-up less than 12 months and with second tumors, was calculated in the three groups of patients; The OS in normal weight plus underweight patients (blue line) was 96% whereas in the overweight group (orange line) it was 93% (*p* = 0.027, HR 3.69, 95%CI = 1.17–13.4), and in the obese group (red line) it was 96% (*p* = 0.169, HR 2.45, 95%CI = 0.68–8.78) (**A**). The PFS was equal to 94% in the normal/underweight group (blue line), 93% in the overweight (orange line), and obese patients (red line). No statistically significant difference was seen amongst the groups (**B**).

**Table 1 cancers-12-01709-t001:** Patients’ characteristics at baseline according to BMI groups.

	All (n = 430)	Underweight (n= 3)	Normal (n = 100)	Overweight (n = 167)	Obese (n = 160)	*p*-Value *
**Age (mean ± SD)**	52.6 ± 23.4	56.7 ± 4.9	48.5 ± 9.3	53.1 ± 9.6	54.6 ± 10	0.032
**Stage at diagnosis N (%)**						0.619
In situ	18 (4.2)	0	3 (3.0)	5 (3.0)	10 (6.2)	
I	221 (51.4)	1 (33.3)	56 (56.0)	91 (54.5)	73 (45.6)	
II	128 (29.8)	1 (33.3)	24 (24.0)	49 (29.3)	54 (33.8)	
III	48 (11.1)	0	14 (14.0)	16 (9.6)	18 (11.3)	
IV	10 (2.3)	0	0	6 (3.6)	4 (2.5)	
nd	5 (1.2)	1 (33.4)	3 (3.0)	0	1 (0.6)	
**Therapy N (%)**						0.639
CT only	29 (6.7)	0	7 (7.0)	11 (6.6)	11 (6.9)	
HT only	200 (46.5)	1 (33.3)	50 (50.0)	77 (46.1)	72 (45.0)	
CT+HT	181 (42.1)	1 (33.3)	39 (39.0)	74 (44.3)	67 (41.9)	
None	14 (3.5)	0	1 (1.0)	5 (3.0)	8 (5.0)	
nd	6 (1.6)	1 (33.4)	3 (3.0)	0	2 (1.2)	
**Type of HT N (%)**						0.041
TAM	130 (34.2)	0	41 (46.6)	51 (34.0)	38 (27.1)	
AI	175 (45.9)	2 (66.7)	29 (33.0)	72 (48.0)	72 (51.4)	
TAM+AI	66 (17.3)	0	15 (17.0)	25 (16.7)	26 (18.6)	
nd	10 (2.6)	1 (33.3)	3 (3.4)	2 (1.3)	4 (2.9)	
**Menopausal status N (%)**						<0.001
Premenopausal	182 (42.3)	0	58 (58.0)	71 (42.5)	53 (33.1)	
Post-menopausal	214 (49.8)	2 (66.7)	34 (34.0)	87 (52.1)	91 (56.9)	
nd	34 (7.9)	1 (33.3)	8 (8.0)	9 (5.4)	16 (10.0)	
**Phenotype N (%)**						0.001
In situ	18 (4.2)	0	3 (3.0)	6 (3.6)	9 (5.6)	
Luminal A	189 (43.9)	0	48 (48.0)	63 (37.7)	78 (48.7)	
Luminal B	124 (28.9)	0	24 (24.0)	60 (35.9)	40 (25.0)	
Luminal/HER2	55 (12.8)	2 (66.7)	12 (12.0)	24 (14.4)	17 (10.6)	
HER2 enriched	11 (2.6)	0	2 (2.0)	5 (3.0)	4 (2.5)	
TNBC	29 (6.7)	0	9 (9.0)	9 (5.4)	11 (6.9)	
nd	4 (0.9)	1 (33.3)	2 (2.0)	0	1 (0.7)	
**Second BC N (%)**	34 (7.9)	0	8 (23.5)	17 (50.0)	9 (26.5)	<0.001
**Size invasive T** **N (%)**						0.001
≤2 cm	250 (60.9)	1 (33.3)	83 (85.6)	120 (74.1)	46 (30.7)	
>2 cm and ≤5 cm	90 (21.8)	0	11 (12.2)	36 (22.2)	43 (28.7)	
>5 cm	45 (10.9)	0	0	0	45 (30.0)	
nd	27 (6.4)	2 (66.7)	3 (2.2)	6 (3.7)	16 (10.6)	
**Histology invasive BC N(%)**						0.71
Ductal	354 (85.9)	3 (100)	82 (84.5)	143 (88.3)	126 (84.0)	
Lobular	55 (13.4)	0	15 (15.5)	17 (10.5)	23 (15.3)	
nd	3 (0.7)	0	0	2 (1.2)	1 (0.7)	
**Grade invasive BC N (%)**						<0.001
I	74 (18.0)	0	32 (33.0)	33 (20.4)	9 (6.0)	
II	156 (37.9)	1 (33.3)	34 (35.1)	69 (42.6)	52 (34.7)	
III	159 (38.5)	0	25 (24.7)	54 (33.3)	80 (60.0)	
nd	23 (5.6)	2 (66.7)	7 (7.2)	6 (3.7)	8 (5.3)	

Abbreviations: SD = standard deviation; Is = in situ carcinoma; nd = not determined; CT = chemotherapy; HT = hormonal therapy; TAM = tamoxifene; AI = aromatase inhibitors; TNBC = triple-negative breast cancer; BC = breast cancer. * The ANOVA test was used to determine differences in clinicopathological features among groups.

**Table 2 cancers-12-01709-t002:** Univariate analysis HR (95% CI) of OS and PFS.

		OS			PFS	
Characteristics	HR	*p* value	95% CI	HR	*p* value	95% CI
BMI						
Normal+UW	1.00 (ref.)			1.00 (ref.)		
Overweight	2.85	0.062	0.95–8.61	2.15	0.085	0.90–5.14
Obese	2.23	0.168	0.71–6.95	1.55	0.348	0.62–3.87
Menopausal status						
Premenopausal	1.00 (ref.)			1.00 (ref.)		
Postmenopausal	1.58	0.399	0.55–4.56	0.94	0.885	0.41–2.16
HT						
TAM	1.00 (ref.)			1.00 (ref.)		
AI	1.44	0.487	0.52–3.99	1.16	0.749	0.47–2.88
TAM+AI	0.83	0.771	0.23–2.97	0.55	0.328	0.17–1.82
Phenotype						
Luminal A	1.00 (ref.)			1.00 (ref.)		
Luminal B	1.22	0.702	0.47–3.42	1.19	0.720	0.46–3.03
Luminal/HER2	1.64	0.183	0.79–3.98	1.74	0.098	0.90–3.33
HER2 enriched	0.88	0.934	0.20–3.7	0.60	0.624	0.32–2.4
TNBC	1.35	0.326	0.72–2.99	1.21	0.124	0.88–3.5
Previous BC	2.78	0.007	1.32–5.86	3.10	0.002	1.54–6.25
Tumor size						
T ≤ 2 cm	1.00 (ref.)			1.00 (ref.)		
T > 2 cm	2.08	0.026	1.09–3.95	1.73	0.079	0.94–3.21
Stage						
0-I	1.00 (ref.)			1.00 (ref.)		
II	0.41	0.417	0.05–3.52	1.46	0.251	0.76–2.81
III-IV	6.29	0.002	2.00–19.8	3.15	0.001	1.60–6.22
Grading						
I	1.00 (ref.)			1.00 (ref.)		
II-III	1.78	0.258	0.81–4.33	1.82	0.074	0.96–4.55

Abbreviations: HR = hazard ratio; CI = confidence interval; OS = overall survival; PFS = progression-free survival; BMI = body mass index; UW = underweight; Ref. = reference; HT = hormonal therapy; TAM = Tamoxifene; AI = aromatase inhibitors; HER2 = human epidermal growth factor receptor 2; TNBC = triple-negative breast cancer; BC = breast cancer; T = tumor size. The Cox regression model was used to calculate hazard ratios

**Table 3 cancers-12-01709-t003:** Multivariate Analysis HR (95% CI) of OS.

		OS	
Characteristics	HR	*p* value	95% CI
BMI			
Normal and UW	1.00 (ref.)		
Overweight	4.57	0.005	1.57–13.4
Obese	2.77	0.072	0.91–8.39
Stage			
I	1.00 (ref.)		
II	1.25	0.583	0.57–2.74
III–IV	4.94	<0.001	2.46–9.92
Second Tumor			
No	1.00 (ref.)		
Yes	4.49	<0.001	1.98–10.2
T ≤ 2 cm	1.00 (ref.)		
T > 2 cm	2.04	0.031	1.07–3.88

Abbreviations: HR = hazard ratio; CI = confidence interval; OS = overall survival; BMI = body mass index; Ref = reference; UW = underweight. T = tumor size. The Cox regression model was used to calculate hazard ratios.

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
