# Peer review of "Lifestyle Intervention on Body Weight and Physical Activity in Patients with Breast Cancer Can Reduce the Risk of Death in Obese Women: The EMILI Study"

_cancers, 2020, doi:10.3390/cancers12071709_

Round 1

Reviewer 1 Report

The manuscript by Cortesi and al. described a single arm intervention study (the EMILI study) with the aim to test the feasibility and efficacy of a lifestyle intervention, based on personalized program of nutritional education, created on the MedDiet, and physical activity, on body weight in 430 Italian patients surgically treated for breast cancer resection. The authors found a BMI reduction in overweight and obese patients. Moreover, analyzing patients’ prognosis they found that obese patients have similar OS and PFS of overweight patients.

Even though the topic is timely and demonstration of the efficacy of specific lifestyle intervention on weight loss could be very interesting and helpful in future trials, the study suffers from several deficiencies that affect the interest level in the manuscript and its relevance for publication in this version.

  • Regarding the primary endpoint of the study (feasibility and efficacy of the intervention), more data and details need to be shown to support conclusions. Which is the intervention compliance? Do all patients benefit from the lifestyle intervention? Which is the percentage of body weight decrease from baseline to the last follow-up in each group, according to BMI? Which is the median time on treatment in each group? The authors stated that the median follow-up is 73 months and the range is 1-124 months. This means that some patients in the cohort haven’t started the protocol yet. Please include in the analyses only patients with at least 1-year follow-up (to have at least the first re-evaluation). Which is the time of enrollment? Had all the patients finished the treatments? The evaluation of the lifestyle intervention efficacy was based only on BMI. How the dietician evaluated the adherence to the dietary recommendation? Through food frequency questionnaires? Please give details in the text. Changes in total physical activity levels should be described also for patient’s groups (according to BMI). The increment in h/week could be plotted.
  • Regarding the survival analysis, the trial design (single arm) does not allow to obtain definitive conclusion on the association between weight loss and prognosis due to an inability to distinguish between the effect of the intervention and of the natural history of the tumors. Indeed, other factors (i.e. tumor characteristics) could have influenced OS in obese and overweight patients. No conclusion on this part can be drawn.

Several minor concerns are also present:

- Data presentation should be improved. The mean alone does not describe the variability of the variable in the cohort analyzed. Please use median and confidence interval in all reported data for better interpretation of results. No mention on the statistical test used to calculate p-value is present in figure/table legends. Is the test used in figure 1 paired? It should be better to plot body weights; in this way data interpretation on the impact of lifestyle interventions is easier.

- Details on definition of BC subtypes and cut-off used for each marker are lacking in M&M.

- Conclusion “…contributing to improve the OS in obese patients compared to overweight ones.” is speculative and not supported by results.

- Follow-up time is not clear. Please decide one follow-up time to give details on OS and PFS. Once it is at 5 years (line 115) and then at 73 months (figure 3).

- Table 1 is unreadable and lacks description of abbreviation (nd). Which is the patients’ age according to BMI subgroups? Distribution of patients with previous BC was not specified.

- Figure 1. It is better to change the graph type to boxplot.

- Figure 3. Panel B is not included in figure legend. Please add the number of patient at risk during the 10 years follow-up.

- Line 123. Interaction analysis between BMI category and age is not clear. Please explain the choice to use 55 years of age as cut-off. Which is the survival for the other two categories?

- Line 141. Change OFS into PFS

- Line 278. Please define the unit for workload > 20 METs/h

-Line 266-269. It is not clear what did authors mean as ‘motivational method used to increase patient physical activity’. Please add further details.

- English editing is required

Reviewer 2 Report

This paper examines the effects of lifestyle changes on breast cancer  recurrence in women with varying BMI.  They have an extensive number of participants in the study and have followed the participants for an extensive number of years.  The data are interesting, interpreted appropriately and offer a little bit of insight into non-medical interventions for breast cancer recurrence. 

Specific comments:

  1. There are a number of small grammatical errors due to non English speakers writing the paper.  I  have listed  those  that I noticed below but may not  be all.  This reviewer suggest a careful reading for this type of edit.
    1. L12 - should read "a sedentary lifestyle"
    2. L47 - should read "being overweight"
    3. L96 - should read "mean decrease"
    4. L113 - remove the word "were"
    5. L114 - instead of  "deceased for"  it should read  "died from"  
    6. L117 - instead of "were joined with" it should  read " were combined with"
    7. L163 - instead of  "in spite of  Mediterranean diet using" it should read  "despite following the Mediterranean diet "
    8. L167 - instead of  "differently from" it should  read "in contrast to"
    9. L173 - it  currently reads "obesity anticipates  the hormonal treatment" This  is an incorrect usage of anticipates and this reviewer is  not sure what the message is for this sentence.  Must be an issue with translation but I don't know what to suggest to adjust it
    10. L185 - instead of "a  correct lifestyle" it should read "a healthy  lifestyle"
    11. L192 - it  currently reads "another interesting  data"  but this isn't a phrase  used in English.  Perhaps the authors mean "It could also be that" but this reviewer is not sure if that changes the authors meaning.
    12. L197 - instead of "along all" it should read "throughout"
    13. L198 - instead of "differently" it should read "in contrast to"
    14. L260 -  insert the word "are" between 'concept' and 'represented'
  2. There were two small issues of spacing - L167 and L183.  
  3. This reviewer requests small  changes  to the figures
    1. In Figure 1 the x axis needs a label
    2. In Figure 1 the figure legend should  state which type of  statistic was  used  to  compute  significance
    3. In Figure 1 the authors should add bars to indicate which groups are being compared and what is significant compared to what.   
    4. Table  1 is very hard to read as it is presented in the preview.  It  could be that this is a formatting issue however the formatting presented to this reviewer was flawed.  There were issues  with  numbers  lining up with  the terms.  If the  table is  moving on  to  two  pages  there should  be a heading on each page  
  4. In  the description of Figure 1  in the results section the authors should say what is compared in the figure.  It is hard to interpret the data as it is currently written.
  5. Concerns with the data:
    1. Figure 1 indicates a significant difference between individuals but are the differences meaningful?  Would a change in BMI like that seen by the patients in the study result in any health benefits?  Is it enough to reduce blood  pressure,  diabetes or  diabetes risk, etc.   This reviewer suggests that  the  authors  address the  concrete benefits of reducing  BMI in the levels seen  by in their patients.
    2. The authors state that  there  is a difference in their study on L167. Could they either point out why there is a difference or speculate  about this for the reader?
    3. While this reviewer knows that the authors collected data for a  substantial amount of time and that  one  has  to  stop collecting data at a certain point could they perhaps speculate what they anticipate if they continued  collecting for more years. 
    4. Could the authors discuss how they address patient compliance with the diet and PA?
    5. Could the authors discuss how the patients felt about the lifestyle changes?  Were they easily adopted by the patients?     

Reviewer 3 Report

Authors present a work addressing the lifestyle intervention on body weight and physical activity in patients with breast cancer can reduce the risk of death in obese women: the EMILI Study. The authors aimed to test the feasibility and efficacy of a lifestyle intervention through diet and PA on body weight in a population of more than 400 patients with BC followed at Modena Cancer Center, Modena, Italy.The general conclusion demonstrates a lifestyle intervention can lead to clinically meaningful weight loss and increase PA in patients with BC, contributing to improve the OS in obese patients compared to overweight ones. The worst prognosis of overweight/obese patients aged more than 55 years is likely related to death from other causes.The manuscript is interesting but demonstrates major and minor issues.

The manuscript in many places is carelessly written and it requires major language correction.

Major comments

  1. The introduction should be extended in respect to heterogeneous nature of breast cancer.
  2. I believe that the authors should provide the limitation of the study.
  3. Statistical methods should be presented in more widen perspective, i.e. which program Authors were used,? etc.
  4. Under table 1 and 2, Authors should extend the abbreviation list including SD, IS, CT, HT, HT, TAM, AI TAM+AI, nd, HER2, TNBC, etc.
  5. Generally we use BMI unit as kg/m2not Kg/m2
  6. This word Hystotype is not English word.
  7. In my knowledge luminal or triple negative subtypes of breast cancer (Table 1) are not a histological subtypes. There are named molecular or intrinsic or biological types. Since, the subtypes determination is based on expression of hormone receptors, human epidermal growth factor receptor 2 (HER2) and expression of proliferation marker-Ki67.
  8. The Discussion section should be extended and focussed on manuscript results.
  9. The median follow-up was equal to 73 months, this period of time is enough to conclude the results. However the range 1-124 is doubtful, since in my opinion Authors should exclude patients with one-month follow-up. One month of follow-up is too short, to provide valuable observations.
  10. In patients’ characteristics, there is a lack of information related to tumour size, histological type, grade.

Minor comment

The references are presented inconsistently with journal guidelines.

Round 2

Reviewer 1 Report

The authors have addressed some of the reviewer’s concerns. In details the authors demonstrated that their intervention study is feasible and effective in reducing BMI and increasing PA especially in patients who most need it. However, survival data are not convincing. Moreover, some issues still need to be addressed.

>Line 240: To state that :”However,  the  worst  prognosis  evidenced  for  overweight  patients  was not  attributable  to the different  stage at the BC onset, but the maintenance of high BMI correlates with an  increased risk of death” a multivariate analysis including BMI and stage should be performed. Otherwise, this statement s not supported by data and should be transform to hypothesis

It is possible that worse prognosis observed in overweight patients derived from the high percentage of 2° tumors in this category that the authors described to be the only parameter significantly associated with PFS and OS. The authors have to perform multivariate analysis to demonstrate that the association between BMI (overweight) and poor prognosis is not dependent from this fact. What happen if patients with 2° event are removed from survival analysis?

Which are the other causes of death? It could be relevant to report and analyze it to understand whether the intervention could impact them, if relevant in the obesity disease. And what about BC specific survival?

>The reviewer does not understand the aim of the analysis between BMI and age. Firstly because the age cut-off is quite arbitrary and also because the authors stated that the majority of death for other causes are in the category OW and OB>55 years. Thus results are not relevant for the study aim. Even results are difficult to understand. Which are the group they compared for each HR reported below:

(HR=2.94, P=0.044, 95% CI 1.03-8.35): OW and OB >55 years vs <55?

(HR=0.57, P=0.616, 95% CI=01.0-5.11): N and UW >55 years vs <55?

(HR=1.32, P=0.636, 95% CI=0.42-4.14): OW and OB <55 vs N and UW <55??

>If the endpoint of the study is to determine the efficacy of the intervention study, 7 patients should be excluded by all analyses not just from the calculation of the median follow-up.

>95% CI has a strange order of magnitude. For example in the abstract: Mean  BMI  decrease  from  baseline to  the  end of  the  study  was equal  to  2.9% (P  = 0.065, 95%CI=  - 1.1_+1.42) in overweight  patients and 3.3% in obese patients  (P=0.048, 95%CI=  -1.82  -0.01). The CI range should contain the median/mean value. What does -1.82 and -0.01 mean referred to 3.3% decrease in BMI? There is something wrong in the calculation of all 95 CI. The 95 % CI of the mean or of the median should be reported not of the p-value.

Line 114: 26.65 kg/m2 (CI=82.5±12.8). Is the upper limit 82.5 and the lower 12.8? If yes please remove the plus/minus symbol

>Line 114-130: the new part is unreadable. It is a copy of the figure. Please insert only number of relevant comparisons.

>Line 142: (211±20.5), there was a 1.5 hours per  week  (124±15.7). What are the numbers in brackets?

>Is the t-test paired or unpaired? The readers have to understand how the p-values reported in the figures or in the table are calculated. Please give details when t-test or anova was used. In figure 1 paired t-test should be used since you followed the same patients over time.

>In figure 1 and 2 It would be easier for readers to see the comparison from which you obtained the p-value without reading the figure legend. Please add a bar under the p-value that link the boxes of the comparison

>Figure 1 legend: “whereas in the different groups of patients the same difference of BMI was seen”. What does it mean? Which kind of box plot has been used? Please specify it to make the reader understand what the dots represent.

>Line 326: Please add a minus sign after HER2 “triple  negative  tumors  were  ER-/PgR-/HER2”

>The formatting problem of table 1 is solved but it is still difficult to read. Please use column header (All, underweight…) Moreover, is quite difficult to understand what percentages are referred to. In one column is relative to the whole population, in the other to the total number of patients in that category. I would suggest to create a new Table 1 in this way:

All (n=430)

Underweight (n=)

Over

Obese

p-value

Age (mean and range)

xx

xx

xx

xx

xx

Stage

xx

I

18 (4.2)

xx

xx

xx

II

xx

xx

xx

xx

III

xx

xx

xx

xx

Type of HT

xx

TAM

130 (34.2)

Or 130/381 (34.2)

AI

Numbers in the table are not consistent. For example:  From the therapy N box the number of patients who received HT are 200+175= 375. But in the box type of HT the total number are 381. The same errors were found comparing the box phenotype and size and grade. Please verify all numbers. Add the type of statistics used to calculate each p value using * or other symbol and describing under the table

>patients at risk in figure 4 should be reported for each category

Reviewer 3 Report

I am appreciate that Authors have introduced all my suggestions. I suggest to publish this article in current form.

Best wishes for you!

Author Response

Thanks

Round 3

Reviewer 1 Report

The authors have addressed most of the reviewer’s concerns.  However, data are poorly presented.

  • To make the abstract flowing I would remove CI and the sentence (bold) to add results on PA: “Mean BMI decrease from baseline to the end of the study was equal to 2.9% (P = 0.065, estimated - 0.45,95%CI= -1.1- 0.54) in overweight patients and 3% in obese patients (P=0.048, estimated -0.92,  95%CI= -1.82_-0.01). After 70  months of follow-up, excluding patients with less than 12 months of  follow-up and patients with second tumors, the 5-year overall survival (OS) was 96%, 96% and 93%,  respectively in normal, obese and overweight patients  (P=0.027, 95%CI=1.82-4.53,  between normal  and  overweight,  P=0.155, 95%CI=0.9-1.4  between  normal  and  obese)” PA…

P=0.027, 95%CI=1.82-4.53,  between normal  and  overweight: data showed in this way make no sense.  Which is the variable? Please rewrite and add HR. OW patients have significantly worse OS than normal HR= 95%CI=1.82-4.53 p=0.027…

Data are not consistent with those in the text (line 214-215 and line 179-181)

  • I suggest making explicit which kind of variable ‘estimated’ is. I guess it is a mean difference. Line 110: with a difference of 6.5% (P<0.001, mean difference: -1.84, 95%CI= -2.75_-0.93). Please add in M&M details on ANOVA repeated measure post test used to compare data at different time point (2 by 2).
  • Weight, BMI and PA section: I suggest choosing to report only one between percent of BMI or weight reduction. This is because the information is the same (using weight is more direct and easier to understand) but % are different (line 123: .. median change of BMI… and to 4.7% (P=0.004, estimated -0.72, 95%CI= -1.20 -0.23) after 12 months in the overweight patients. …….line 129: Overweight women moved from 72.2 kg, at the entry to 67.6 after one year ( - 6.3% P<0.001, estimated -1.90, 95%CI= -3.10 -0.70), thus it is quite hard to read and follow it.
  • Line 128: please check and correct: 3.2%, P=0.048, estimated -1.51, estimated -0.91, 95%CI= -1.80  -0.01
  • Line 165: Please check and correct. The estimated difference between BMI, weight and physical activity along time-points was performed by  means  of  ANOVA  with  repeated  Figure is related to PA.
  • Line 110 which test was used to compared patient ages? I don’t think it is chi square test.
  • Line 174: BC specific survival in the overall cohort (Figure 4) is not necessary if then the analysis according to BMI (figure 5) is not presented also for BC specific survival. I guess the authors did not insert it because no differences emerged between OW and Normal. One should declares it in the text (also data not shown) and discuss it. Figure 4 can be deleted.
  • Line 185: The sentence is still not clear. “55 years was the  cut-off  by  which  underweight  and  normal  weight  women  had  no  incremental  risk  of  death,  whereas overweight and obese women aged ≥55 years, showed a statistically significant increase of  death (HR=2.94, P=0.044, 95% CI 1.03-8.35) compared to normal and underweight patients  aged less  than 55 years, whereas no differences were seen in comparison with normal and underweight people  aged ≥55 years (HR=0.57, P=0.616, 95% CI=01.0-5.11)”.

Incremental vs who? What do you mean for interaction? (line 183). Did you perform an interaction test in multivariate analysis? I still do not understand this analysis and the results. Why do you think it is so important that OB/OW patients aged>55 have worse prognosis than N/UW aged<55? I guess this significant difference could derive from other characteristics because the two compared groups are so different. Please discuss it or delete the analysis.

  • Line 224. Data are not consistent to the table. Please check and correct
  • Line 234: As shown, overweight patients, but not obese ones, advanced disease stages (III… please change the verb.
  • Univariate analysis should be performed for all variables analyzed (stage, T and grade.. need to be added)
  • Multivariate analysis demonstrated that OW patients have significantly worse prognosis independently from stage and the presence of previous tumor. I would also insert in multivariate other variables that result significantly associated with OS in univariate analyses, if they will be associated (T or grade).
  • Line 271-274: check English.
